# Triamidoamine thorium-arsenic complexes with parent arsenide, arsinidiide and arsenido structural motifs

Elizabeth P. Wildman[1], Gábor Balázs[2], Ashley J. Wooles[1], Manfred Scheer[2] & Stephen T. Liddle[1]

Despite a major expansion of uranium–ligand multiple bond chemistry in recent years, analogous complexes involving other actinides (An) remain scarce. For thorium, under ambient conditions only a few multiple bonds to carbon, nitrogen, phosphorus and chalcogenides are reported, and none to arsenic are known; indeed only two complexes with thorium–arsenic single bonds have been structurally authenticated, reflecting the challenges of stabilizing polar linkages at the large thorium ion. Here, we report thorium parent–arsenide (ThAsH$_2$), –arsinidiides (ThAs(H)K and ThAs(H)Th) and arsenido (ThAsTh) linkages stabilized by a bulky triamidoamine ligand. The ThAs(H)K and ThAsTh linkages exhibit polarized-covalent thorium–arsenic multiple bonding interactions, hitherto restricted to cryogenic matrix isolation experiments, and the AnAs(H)An and AnAsAn linkages reported here have no precedent in f-block chemistry. 7s, 6d and 5f orbital contributions to the Th–As bonds are suggested by quantum chemical calculations, and their compositions unexpectedly appear to be tensioned differently compared to phosphorus congeners.

[1] School of Chemistry, The University of Manchester, Oxford Road, Manchester M13 9PL, UK. [2] Institute of Inorganic Chemistry, University of Regensburg, Universitätsstr.31, Regensburg 93053, Germany. Correspondence and requests for materials should be addressed to M.S. (email: manfred.scheer@ur.de) or to S.T.L. (email: steve.liddle@manchester.ac.uk).

Despite the developed nature of uranium–ligand multiple bond chemistry[1–4]—that is burgeoning due to a pressing need to improve our fundamental understanding of the chemical bonding of actinide (An) elements[5–9] and how this might impact reactivity and nuclear waste clean-up[10,11]—the chemistry of other An–ligand multiple bonds, aside from actinyls, $AnO_2^{n+}$ (ref. 12), remains extremely rare[13]. However, in order to truly delineate bonding trends across the actinides, and thus draw inferences as to their competitive bonding and reactivity, comparisons between structurally analogous complexes that contain different actinide metals are required, but concerted comparisons of such families of compounds remain relatively rare[14–21] due to a paucity of synthetic methodologies to prepare them or the radiological limitations encountered in this area. In this regard, thorium is an attractive element to study, both in terms of radiological practicality and also the nature of its bonding when compared to uranium, that is, 6d versus 5f character[19,21–24], respectively.

Thorium–ligand multiple bond chemistry is an attractive area for studying chemical bonding because of the intrinsic features of metal–ligand multiple bonds, but it is far less developed than uranium[1–4], being restricted to a few carbene[21,25–30], imido[31–33] and chalcogenido[22–24,34–36] complexes. Very recently the first thorium–phosphinidene and µ-phosphido complexes under ambient conditions[37] added to the preparation of a thorium–phosphido complex in frozen argon matrix isolation conditions[38]. The bonding contrast between uranium and thorium is usually framed in terms of computationally determined straight trade-offs between 5f and 6d orbital character[23], but our recent work on Th–P bonds[37] showed an unexpected intrusion of 7s orbital character into the bonding, and notably this effectively came at the expense of 6d and not 5f orbital participation. In order to ascertain whether this is a unique or more broadly applicable phenomenon would require comparison to closely related compounds and in that regard arsenic congeners are the logical next step. However, reflecting the cumulative instability of pairing a large, polar metal like thorium with increasingly large, soft p-block elements[39–47] presents an increasing synthetic challenge regarding their preparation and isolation. Indeed, examples of

thorium complexes with bonds to pnictide (Pn) elements heavier than phosphorus are exceedingly rare, and under ambient conditions the only two structurally characterized thorium–arsenic complexes are the Zintl-polyarsenide [{Th($\eta^5$-1,3-Bu$^t_2$C$_5$H$_3$)$_2$}$_2$ (µ-$\eta^3$:$\eta^3$-As$_6$)] (ref. 48) and the bis(arsenide) [Th($\eta^5$-C$_5$Me$_5$)$_2$ (AsH-2,4,6-Pr$^i_3$C$_6$H$_2$)$_2$] (ref. 39), which are principally Th–As σ-bonded species. In contrast, thorium–arsenic multiple bonded compounds are restricted to F$_3$Th≡As formed under cryogenic matrix isolation conditions[38]. Thus, there are very few thorium–arsenic bonds, and none with polarized-covalent multiple bonding interactions under ambient conditions. More widely, even multiple bonds between transition metals and arsenic remain quite rare[49–55].

Previously, following on from our uranium-nitride work[56–59] we have reported the synthesis of terminal parent uranium–amide, –phosphanide, and –arsenide complexes [U(Tren$^{TIPS}$)(EH$_2$)] [Tren$^{TIPS}$ = N(CH$_2$CH$_2$NSiPr$^i_3$)$_3$; E = N, P, As], the corresponding parent uranium–imido, –phosphinidene and –arsinidene complexes [U(Tren$^{TIPS}$)(EH)][K(L)$_2$] (L = a 15-crown-5 ether), and the arsenido tetramer [U(Tren$^{TIPS}$)AsK$_2$]$_4$ (refs 60–62). More recently we extended this chemistry to thorium and prepared parent phosphanide, phosphinidene, phosphinidiide and µ-phosphido analogues[37], and noted that the increase of single bond covalent radii of thorium (1.75 Å) compared to uranium (1.70 Å) (ref. 63) resulted in the thorium derivatives being on the cusp of stability compared to uranium analogues; this prompted us to consider the preparation of the thorium–arsenic analogues since according to Pyykkö arsenic is ≥0.1 Å larger than phosphorus[63], and thus represents a significant synthetic challenge to be stabilized at thorium from a steric perspective in addition to aforementioned bond polarity and covalency arguments.

Here, we report the synthesis and characterization of thorium–arsenic complexes with ThAsH$_2$ parent–arsenide, ThAs(H)K and ThAs(H)Th parent–arsinidiide and ThAsTh µ-arsenido linkages stabilized by the Tren$^{TIPS}$ ligand[59]. The thorium–arsenic linkages in this study are the most ionic of our An–Pn combinations to date, but quantum chemical calculations suggest that both ThAsH$_2$ and ThAs(H)K display modest covalency. This provides an opportunity to compare to previously reported

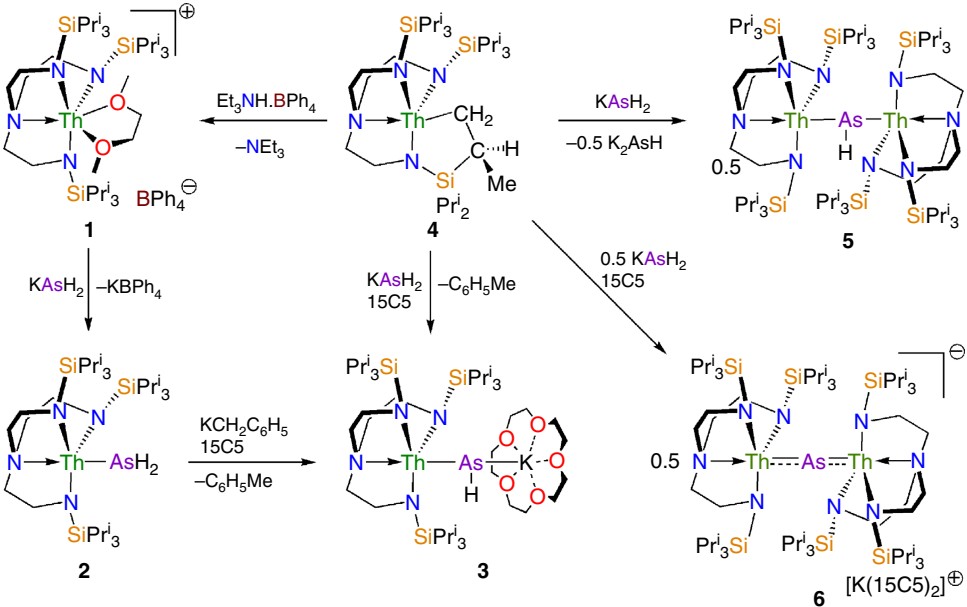

**Figure 1 | Synthetic routes to thorium arsenic complexes.** Complex **2** is prepared by a salt elimination reaction of **1**. Complex **3** is prepared either by deprotonation of **2** or by direct protonation of **4**, both in the presence of crown ether. Complexes **5** and **6** are prepared by protonation of **4** with the reaction outcome being principally determined by the stoichiometric ratio of **4** to KAsH$_2$.

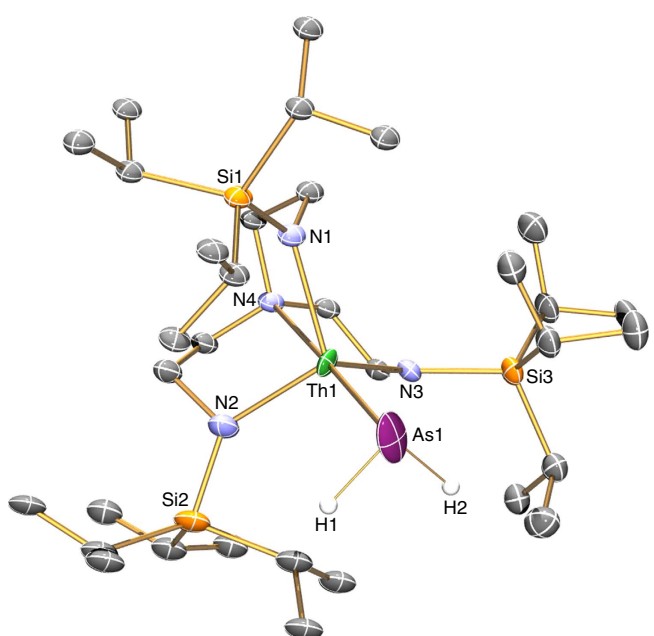

**Figure 2 | Molecular structure of 2 at 150 K with displacement ellipsoids set to 40%.** Non-arsenic-bound hydrogen atoms and minor disorder components are omitted for clarity.

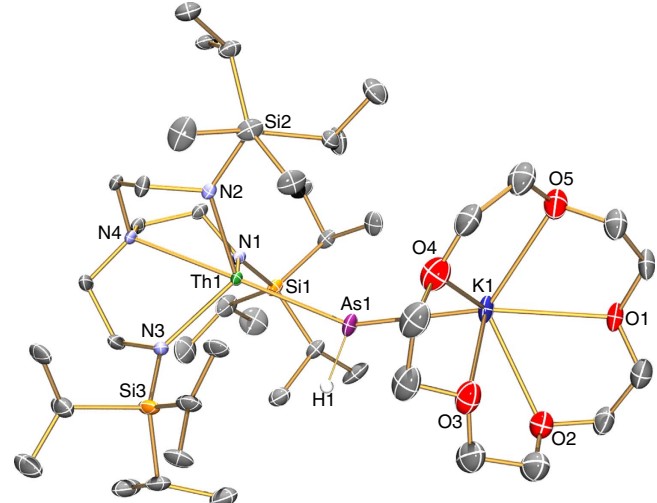

**Figure 3 | Molecular structure of 3 at 150 K with displacement ellipsoids set to 40%.** Non-arsenic-bound hydrogen atoms and minor disorder components are omitted for clarity.

uranium–arsenic, and thorium–phosphorus congeners, revealing unexpected contrasts of the balance and tensioning of the 7s/6d/5f contributions in the Th–As bonds when compared to Th–P congeners[37,62].

## Results

**Synthesis.** Treatment of [Th(Tren$^{TIPS}$)(DME)][BPh$_4$] (**1**) (ref. 37) with KAsH$_2$ (ref. 64) in DME, followed by work-up afforded yellow crystals of the parent arsenide complex [Th(Tren$^{TIPS}$)(AsH$_2$)] (**2**) in 71% yield, Fig. 1. While the Tren$^{TIPS}$ proton environments are clearly observed in the $^1$H NMR spectrum of **2**, the AsH$_2$ hydrogen resonances could not be located. However, the presence of AsH$_2$ in **2** is confirmed by two As–H stretches at 2,051 and 2,064 cm$^{-1}$ in its ATR-IR spectrum, and these stretches compare very well to computed As–H stretches of 2,050 and 2,066 cm$^{-1}$ from an analytical frequencies calculation (Supplementary Methods). Further, when KAsD$_2$ (ref. 37) is used in the synthesis instead of KAsH$_2$ to prepare [Th(Tren$^{TIPS}$)(AsD$_2$)] (**2D**) for isotopic labelling purposes, the absorptions at 2,051 and 2,064 cm$^{-1}$ shift, as expected, into the fingerprint region of the ATR-IR spectrum (Supplementary Figure 1).

When **2** is treated with one equivalent of benzyl potassium/15-crown-5 ether (15C5), the parent arsinidiide complex [Th(Tren$^{TIPS}$)(μ-AsH)K(15C5)] (**3**) is isolated after work-up. Complex **3** can also be accessed directly from KAsH$_2$, 15C5 and the cyclometallate complex [Th{N(CH$_2$CH$_2$NSiPr$^i_3$)$_2$ (CH$_2$CH$_2$NSiPr$^i_2$C[H]MeCH$_2$)}] (**4**) (ref. 19) when reacted in a 1:1:1 ratio. Complex **3** is typically obtained as an analytically clean microcrystalline powder in 70% isolated yield. Like **2**, the AsH resonance of **3** could not be located in the $^1$H NMR spectrum, but a weak and broad As–H stretch is found around 1,930 cm$^{-1}$ in the ATR-IR spectrum, which compares to a computed As–H frequency stretch of 1,974 cm$^{-1}$. To confirm that the ~1,930 cm$^{-1}$ absorption is an As–H stretch we prepared [Th(Tren$^{TIPS}$)(μ-AsD)K(15C5)] (**3D**) and found like **2/2D** that upon deuteration the ~1,930 cm$^{-1}$ band

to be greatly reduced in intensity (Supplementary Figure 2) due to an isotopomer shift into the fingerprint region.

When the 1:1 reaction of **4** with KAsH$_2$ is carried out in the absence of 15C5 the yellow dithorium parent arsinidiide complex [{Th(Tren$^{TIPS}$)}$_2$(μ-AsH)] (**5**) is the only isolable compound from a complex mixture of otherwise intractable decomposition products. Despite the mixture of products, however, **5** is isolated in 74% crystalline yield, Scheme 1, presumably with concomitant elimination of K$_2$AsH, which is a known product from the reaction 2 KAsH$_2$ → K$_2$AsH + AsH$_3$ (ref. 64). As above, the AsH resonance could not be located in the $^1$H NMR spectrum, but the ATR-IR spectrum reveals a broad and weak absorption at 1,922 cm$^{-1}$ that compares well to a computed As–H stretching frequency of 1,987 cm$^{-1}$. Inspection of the ATR-IR spectrum of [{Th(Tren$^{TIPS}$)}$_2$(μ-AsD)] (**5D**) reveals a major reduction in intensity of the 1,922 cm$^{-1}$ absorption, confirming this absorption as an As–H stretch, but the resulting As–D stretch moves well into the fingerprint region and could not be located (Supplementary Figure 3).

Seeking to access a terminal Th = AsH unit by another route to that which resulted in the production of **5**, when two equivalents of 15C5 or one equivalent of 2,2,2-cryptand are used in the deprotonation of **2** with benzyl potassium, unlike the analogous uranium chemistry[62], the terminal arsinidene complex [Th(Tren$^{TIPS}$)(AsH)][K(L)$_2$] (L = crown or cryptand) could not be isolated despite exhaustive efforts. It would seem that the combination of the large thorium and arsenic centres with a weak, polar and labile putative Th = AsH linkage exceeds the steric stabilizing properties of Tren$^{TIPS}$. Instead, with no bulky As-group to kinetically protect the terminal Th = AsH linkage, the dithorium–arsenido complex [{Th(Tren$^{TIPS}$)}$_2$(μ-As)][K(15C5)$_2$] (**6**) is the sole isolable product with elimination of KAsH$_2$ implicated; in effect K(15C5) and Th(Tren$^{TIPS}$) thus act as protective trapping groups to the [(Tren$^{TIPS}$)ThAsH]$^-$ units in **3** and **5**, respectively.

Since the As–H linkages are evidently polar and labile, we sought to prepare **6** rationally, and find that **6** can be accessed via a deliberate synthesis where **4** is reacted with KAsH$_2$ in a 2:1 ratio in the presence of two equivalents of 15C5; this produces **6** in 74% yield as a microcrystalline powder, Scheme 1. The NMR and ATR-IR data for **6** are consistent with its formulation but are not particularly informative given the absence of any key functional group spectroscopic handles.

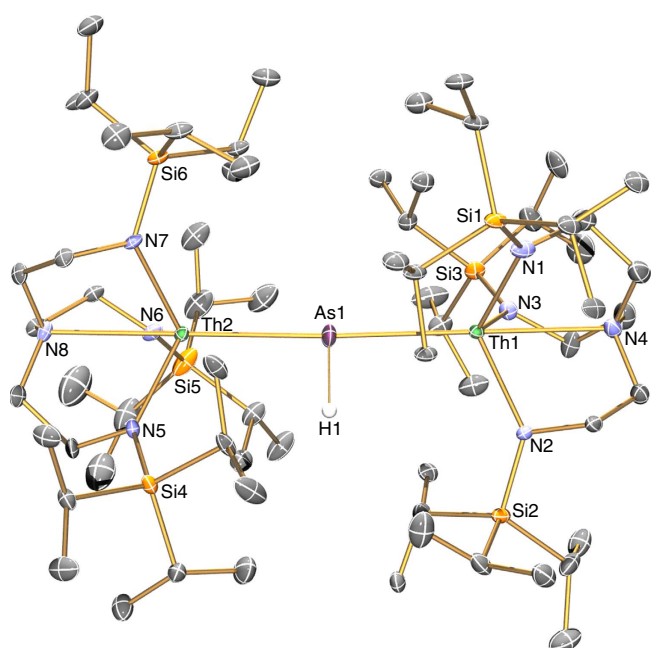

**Figure 4 | Molecular structure of 5 at 150 K with displacement ellipsoids set to 40%.** Non-arsenic-bound hydrogen atoms and minor disorder components are omitted for clarity.

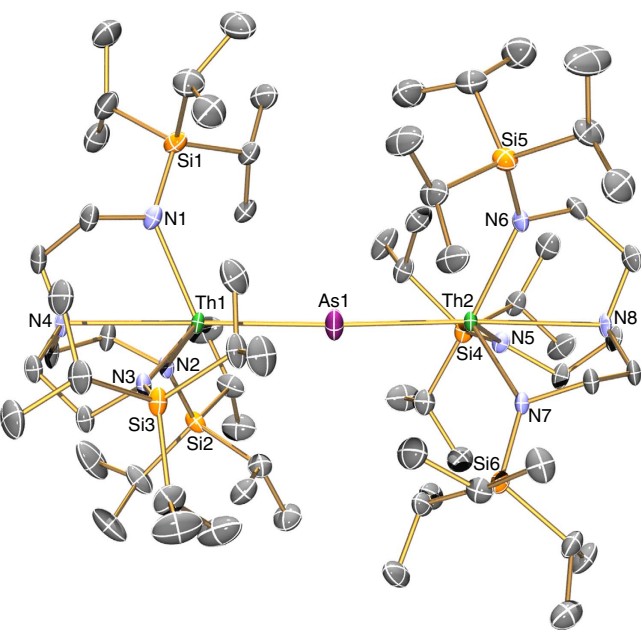

**Figure 5 | Molecular structure of 6 at 150 K with displacement ellipsoids set to 40%.** Hydrogen atoms, minor disorder components and the $[K(15C5)_2]^+$ cation component are omitted for clarity.

**Solid state structures**. The solid state molecular structures of **2**, **3**, **5** and **6** were confirmed by single crystal X-ray diffraction (Figs 2–5). The thorium–arsenic distances in **2**, **3**, **5** and **6** are 3.065(3), 2.8565(7), 2.9619(6)/3.0286(6) and 2.8063(14)/2.8060(14) Å, respectively, and can be compared to additive sum of the single and double bond radii of these elements of 2.96 and 2.57 Å (ref. 63), respectively, and experimentally determined thorium–arsenic single bond distances of 2.913(2)-3.044(2) and 3.0028(6) observed in $[\{Th(\eta^5\text{-}1,3\text{-}Bu^t_2C_5H_3)_2(\mu\text{-}\eta^3\text{:}\eta^3\text{-}As_6]$ (ref. 48) and $[Th(\eta^5\text{-}C_5Me_5)_2(AsH\text{-}2,4,6\text{-}Pr^i_3C_6H_2)_2]$ (ref. 39), respectively. The Th–$N_{amide}$ and –$N_{amine}$ bond lengths are similar to analogous distances in other Th–Tren complexes[19,65], and no obvious trend is discernable with respect to the anionic thorium-components of **3** and **6** versus the neutral formulations of **2** and **5** or the presence of the $(AsH)^{2-}$ and $As^{3-}$ in **3** and **6**, respectively, that might have been anticipated to exhibit *trans*-influences to the coordinated $Tren^{TIPS}$ amines as is the case in the ThPTh counterpart[37].

In **3** the Th–As–H angle was found, by a combination of crystallography coupled to DFT calculations, to be 79.1(2)°, which suggests the presence of a $Th\cdots H$ interaction; overall the arsinidiide centre in **3** is essentially planar $[\sum\angle = 356.77(5)°]$, but asymmetrically arranged due to the obtuse Th–As–K angle $[150.87(5)°]$ that is undoubtedly imposed by the 15C5 ligand. In contrast, the arsinidiide centre in **5** exhibits an essentially planar T-shaped geometry $[\sum\angle = 359.64(6)°]$, but as expected the Th–As–Th angle in **6** is approaching linearity $[\sum\angle = 177.04(6)°]$.

**Computational characterization**. To understand the nature of the thorium–arsenic interactions in **2**, **3**, **5** and **6**, we computed their electronic structures using the full models of **2**, **3** and **5** and the full thorium anion component of **6** (**6⁻**) (Table 1). The geometry-optimized structures reproduce experimental bond lengths and angles to within 0.08 Å and 3° (Supplementary Tables 1–4) and so we conclude that they represent

qualitative descriptions of the electronic structures of these molecules.

The computed thorium MDC-$q$ charges for **2**, **3**, **5** and **6⁻** are in-line with thorium(IV)–Tren complexes[37], and suggest the Th–As linkages in the former pair may be more covalently bonded than those of the latter pair which is consistent with the NBO analyses (see below). The calculated arsenic MDC-$q$ charges for **2**, **3**, **5** and **6⁻** reflect the changes moving from formally monoanionic $(AsH_2)^-$ to dianionic and multiply bonded $(AsH)^{2-}$ to dianionic but bridging $(AsH)^{2-}$ to trianionic and bridging $As^{3-}$, respectively. The thorium–arsenic Mayer bond orders for **2**, **3**, **5** and **6⁻** reflect the formal Th–As single, double, single and overall double bond interactions in these molecules and are also consistent with the arsenic MDC-$q$ charges. For comparison, the Mayer bond orders of the Th–$N_{amide}$, Th–$N_{amine}$ and As–K linkages average 0.70, 0.25 and 0.13, respectively, showing that although the Th–As linkages are polarized those of **3** and **6⁻**, even though bridging, are consistent with multiple bond formulations.

Inspection of the Kohn Sham frontier molecular orbitals for **2**, **3**, **5** and **6⁻** (Figs 6–9) reveal σ-covalent, σ-/π-covalent, σ-covalent-π-dative, and σ-/π-/π-covalent bonding interactions between the thorium and arsenic centres, respectively. Interestingly, although the arsenic 4p-orbitals form pseudo triple bond interactions to each thorium in **6⁻** they are so polarized that they equate to thorium–arsenic double bonds overall in a Lewis bonding scheme. These molecular orbitals as well as arsenic 4p character contain a varying mix of thorium 7s, 7p, 6d and 5f character and although they are reasonably well defined they contain small intrusions of nitrogen lone pair orbital coefficients and so we examined the NBOs of these orbitals to gain a clearer picture of the thorium–arsenic bonding.

For **2** and **3**, NBO calculations suggest a modest degree of covalency (Supplementary Figs 4 and 5), though the bonding is clearly highly polarized. As suggested by the computed thorium MDC-$q$ charges, NBO returns the thorium–arsenic interactions in **5** and **6⁻** as ionic (Supplementary Figs 6 and 7), which means

**Table 1 | Computed data for the thorium-arsenic compounds reported in this study.**

| Entry[‡] | Bond length and index | | Atomic charges | | NBO σ-component[*] | | | NBO π-component[*] | | | QTAIM[†] | | | |
|---|---|---|---|---|---|---|---|---|---|---|---|---|---|---|
| | Th–As[§] | BI[‖] | $q$Th[¶] | $q$As[#] | %Th | %As | Th 7s:7p:6d:5f | %Th | %As | Th 7s:7p:6d:5f | $\rho(\mathbf{r})$ | $\nabla^2\rho(\mathbf{r})$ | $H(\mathbf{r})$ | $\varepsilon(\mathbf{r})$ |
| **2** | 3.133 | 0.68 | 2.56 | − 0.49 | 8 | 92 | 18:2:46:34 | — | — | — | 0.04 | 0.03 | − 0.01 | 0.04 |
| **3** | 2.888 | 1.09 | 2.55 | − 1.47 | 9 | 91 | 8:0:48:44 | 9 | 91 | 0:0:37:63 | 0.06 | 0.06 | − 0.02 | 0.34 |
| **5** | 3.063 | 0.83 | 2.95 | − 2.34 | <5 | >95 | — | <5 | >95 | — | 0.04 | 0.05 | − 0.01 | 0.25 |
| | 3.064 | 0.90 | 3.12 | — | <5 | >95 | — | <5 | >95 | — | 0.04 | 0.05 | − 0.01 | 0.28 |
| **6**[−] | 2.845 | 1.25 | 3.70 | − 2.06 | <5 | >95 | — | <5 | >95 | — | 0.06 | 0.08 | − 0.02 | 0.01 |
| | 2.848 | 1.23 | 3.69 | — | <5 | >95 | — | <5 | >95 | — | 0.06 | 0.08 | − 0.02 | 0.02 |

*Natural Bond Orbital (NBO) analyses; the electron occupancies of these orbitals are ≥97%.
†QTAIM topological electron density [$\rho(\mathbf{r})$], Laplacian [$\nabla^2\rho(\mathbf{r})$], electronic energy density [$H(\mathbf{r})$] and ellipticity [$\varepsilon(\mathbf{r})$] bond critical point data.
‡Molecules geometry optimized without symmetry constraints at the restricted LDA VWN BP TZP/ZORA level.
§Calculated Th-As distances (Å).
‖Mayer bond indices.
¶MDC-$q$ charges on thorium.
#MDC-$q$ charges on arsenic.

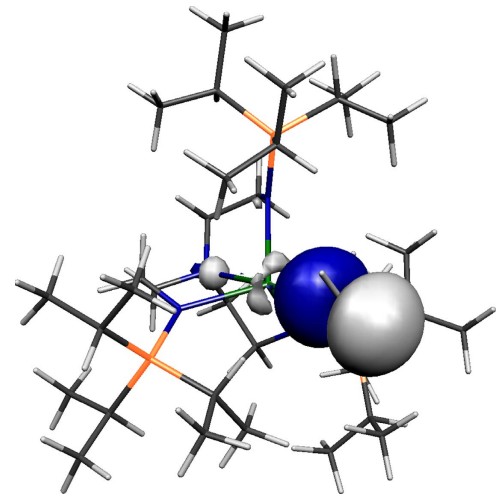

**Figure 6 | Kohn Sham molecular orbital representation of the principal Th–As interaction of 2.** HOMO ( − 1.460 eV) represents the principal thorium–arsenic covalent σ-bonding interaction in **2**.

the thorium contribution is <5%, the NBO cut-off, and the arsenic contributions are >95%.

Inspection of the topological electron density of the thorium–arsenic bonds in **2**, **3**, **5** and **6**[−] identified 3, − 1 critical points in all cases indicating the presence of thorium–arsenic bonds, but in-line with the DFT and NBO orbital-based pictures suggests fairly ionic bonds throughout. We note, however, that the $\rho$ values for **3** and **6**[−] are moderately higher than for **2** and **5**, consistent with their multiple bond character. Additionally, the ellipticity values for **2**, **3**, **5** and **6**[−] show cylindrical bonds for **2** and **6**[−], and asymmetric bonds for **3** and **5** that are consistent with the σ-covalent, σ-/π-covalent, σ-covalent-π-dative, and σ-/π-/π-covalent bonding interactions of **2**, **3**, **5** and **6**[−], respectively, as suggested by the DFT and NBO methods[66].

## Discussion

The synthesis and characterization of thorium parent–arsenide (ThAsH₂), –arsinidiides (ThAs(H)K and ThAs(H)Th) and arsenido (ThAsTh) linkages stabilized by a bulky triamidoamine ligand introduces a new range of An–As bonds. The parent AsH$_n$ linkages are notable for their sterically unencumbered natures at an ion as large as thorium[67–70], and the ThAs(H)K and ThAsTh linkages, even though not terminal, represent the first

thorium–arsenic multiple bonds under ambient conditions. Surprisingly, given recent progress in non-aqueous actinide chemistry, the ThAs(H)Th and ThAsTh linkages reported here have no precedent in f-block chemistry and indeed are rare even in transition metal chemistry[69–75]. All attempts to prepare a terminal parent Th = AsH linkage resulted in the isolation of a compound with a ThAsTh arsenido structural motif, demonstrating that Th = AsH is too fragile to be stabilized even by the very bulky Tren^TIPS ligand, and thus this thorium–arsenic combination surpasses the stabilizing capability of Tren^TIPS highlighting the synthetic challenges of stabilizing bonding between such large metal/metalloidal ions. The ability to prepare complex **3** by more than one method is notable. The same situation was found for the Th–P analogue, but is perhaps more remarkable here due to the increased polarity of the resulting Th–As bond; while greater reactivity could be anticipated, this would be expected to result in less product selectivity so the fact that a well-defined product can be routinely isolated as the majority product is unexpected when considering the number of bond-breaking and -forming steps that occur concurrently to give **3** when prepared from cyclometallate **4**.

The thorium–arsenic single bond in **2** and double bond in **3** can be considered to be long when compared to the additive sum of the single bond radii of thorium and arsenic; this trend has been observed with the related U–P, U–As and Th–P complexes and can be related to the steric demands of the Tren^TIPS ligand and for **3** to the anionic formulation of the Th-component[37,61,62]. The thorium–arsenic bond lengths in **5** are intermediate to **2** and **3**, which reflects the dianionic nature of the (AsH)$^{2-}$ unit coupled to its bridging nature and the presence of two highly polarizing thorium centres in **5** but only one thorium and a softer potassium ion in **3**. The additional charge load of As$^{3-}$ compared to (AsH)$^{2-}$ accounts for the moderately shorter thorium–arsenic bond lengths in **6** compared to **5** and **3** even when two polarizing thorium ions are coordinated to the trianionic arsenido ion, and suggests that a Th = As = Th core with two formal double bond interactions, in a Lewis bonding scheme, can be considered to be present in **6** as is the case with the ThPTh analogue[37]. One notable observation is that in [{Th(Tren^TIPS)}₂(μ-P)]$^-$ the Th–N$_{amine}$ distances are quite long, suggesting a *trans*-influence of the trianionic phosphido ligand; however, an analogous effect is not duplicated in **6**, which perhaps reflects the lower charge density at arsenic compared to phosphorus and the more ionic Th–As bonding compared to Th–P. The Th–As–H angle in **3** of 79.1(2)° suggests, like the Th–P analogue[37], that a Th···H agostic- or anagostic-type interaction may be present, but unlike the Th–P analogue unfortunately no As–H coupling constant data are available from which to judge whether this is actually the case.

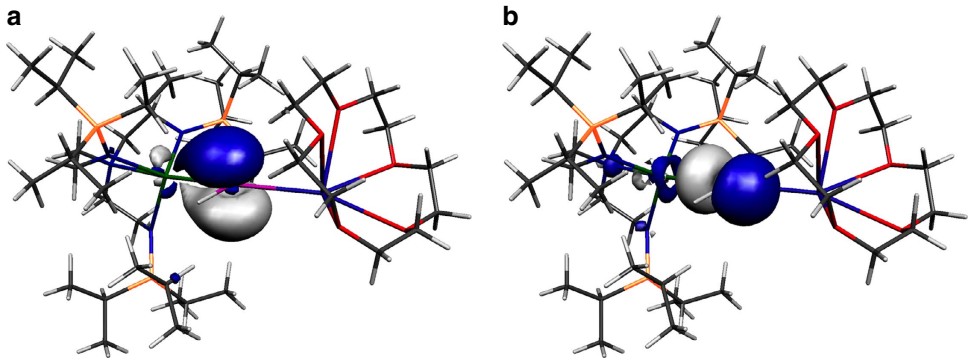

**Figure 7 | Kohn Sham molecular orbital representations of the principal Th–As interactions of 3.** HOMO (**a**, − 2.619 eV) and HOMO-1 (**b**, − 3.032 eV) represent the two principal thorium–phosphorus covalent π- and σ-bonding interactions in the anion component of **3**, respectively.

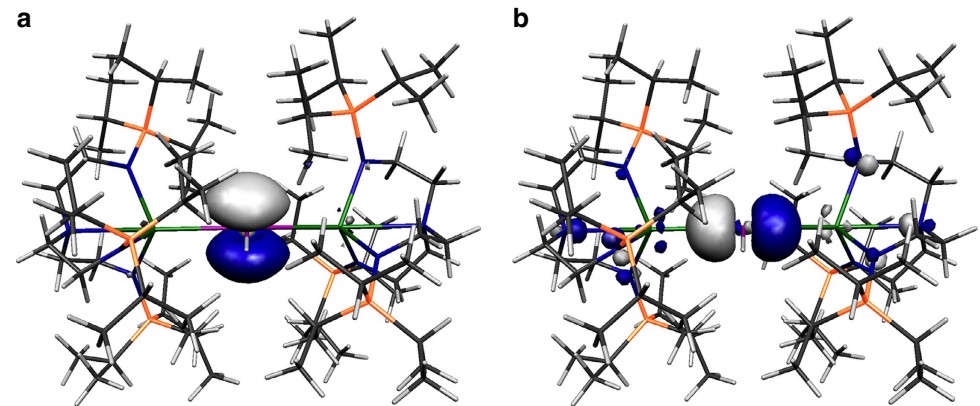

**Figure 8 | Kohn Sham molecular orbital representations of the principal Th–As interactions of 5.** HOMO (**a**, − 3.615 eV) and HOMO-1 (**b**, − 4.372 eV) represent the two principal thorium–arsenic dative π-symmetry and covalent σ-bonding interactions in **5**.

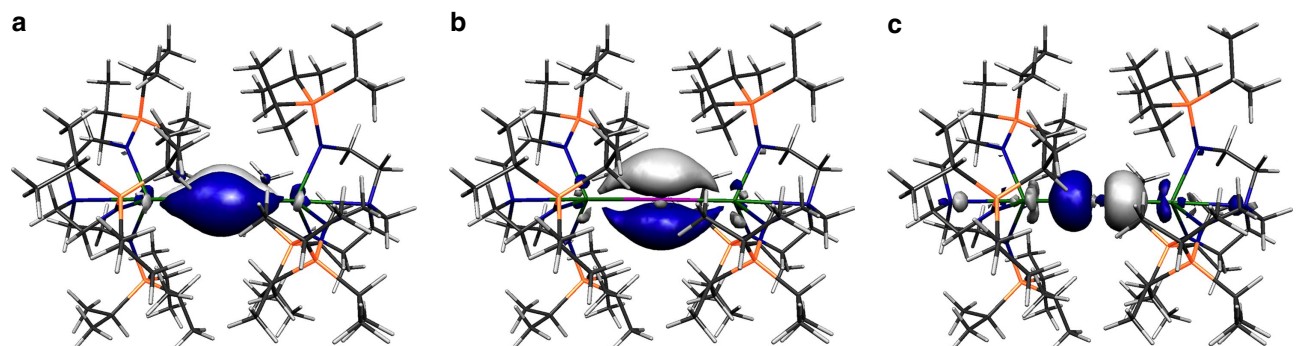

**Figure 9 | Kohn Sham molecular orbital representations of the principal Th–As interactions of 6⁻.** HOMO (**a**, − 0.951 eV), HOMO-1 (**b**, − 0.964 eV) and HOMO-2 (**c**, − 1.486 eV) represent the three principal thorium–arsenic covalent π- and σ-bonding interactions, respectively, in the anion component of **6**. The two Th–As π-interactions are delocalized in the molecular orbital model but together with the σ-bond these pseudo triple bonds equate to Th = As double bonds in a Lewis bonding scheme.

The computational data for **2**, **3**, **5** and **6** are largely in-line with tetravalent uranium– and thorium–Tren complexes, but comparison of **2** and **3** to $[U(Tren^{TIPS})(AsH_2)]$ (**I**) and $[U(Tren^{TIPS})(\mu\text{-}AsH)K(2,2,2\text{-crypt})]$ (**II**) is particularly informative, Table 2 (ref. 62). For **2** and **I**, the An–As Mayer bond orders and MDC-$q$ charges are essentially the same, but the An–As bond in **I** is 13% U compared to 8% Th in **2**. Although the uranium component to the U–As σ-bond in **I** is split 3/2/47/48% 7s/7p/6d/5f the corresponding breakdown in **2** shows similar 6d participation in

both complexes, but in **2** the 5f contribution is depleted due to a 7s component increased 15% compared to **I**. For **3** and **II**, the uranium–arsenic Mayer bond order of 1.35 in **II** is clearly larger than the thorium–arsenic value of 1.09 in **3**, but the uranium and arsenic MDC-$q$ charges of 2.68 and − 1.34 in **II** are similar to the corresponding values in **3**. In **II** the uranium– arsenic σ-bond is 17% U (1/0/48/51% 7s/7p/6d/5f) and 83% As (5/95% 4s/4p) and the π-bond is 22% U (0/1/36/63% 7s/7p/6d/5f) and 78% As (100% 4p). Clearly, the corresponding

**Table 2 | Comparison of computed NBO data for a selection of structurally related actinide-pnictide complexes.**

| Entry[†] | NBO σ-component* | | | NBO π-component* | | |
|---|---|---|---|---|---|---|
| | %An[‡] | %E[§] | An 7s:7p:6d:5f | %An[‡] | %E[§] | An 7s:7p:6d:5f |
| **2** | 8 | 92 | 18:2:46:34 | — | — | — |
| **I** | 13 | 87 | 3:2:47:48 | — | — | — |
| **3** | 9 | 91 | 8:0:48:44 | 9 | 91 | 0:0:37:63 |
| **II** | 17 | 83 | 1:0:48:51 | 22 | 78 | 0:1:36:63 |
| **III** | 8 | 92 | 12:2:38:48 | — | — | — |
| **IV** | 13 | 87 | 2:2:51:45 | — | — | — |
| **V** | 12 | 88 | 4:0:44:52 | 14 | 86 | 0:1:54:45 |
| **VI** | 24 | 76 | 0:0:20:80 | 0:1:30:69 | | |

*Natural Bond Orbital (NBO) analyses; the electron occupancies of these orbitals are ≥97%.
†Molecules geometry optimized without symmetry constraints at the restricted or unrestricted LDA VWN BP TZP/ZORA level.
‡An = thorium or uranium.
§E = P or As.

thorium contributions to the thorium σ- and π-bonds in **3** are diminished by 8–13% compared to the uranium contributions in **II**, and again although the 6d contributions in the σ- and π-bonds are essentially the same for uranium and thorium, in the σ-bond the 5f contribution for thorium is lower due to 7% increased 7s contribution but for the π-bond the 5f contribution is the same for thorium and uranium. Overall, the picture that emerges is that thorium is more ionic than uranium, but for these actinide–arsenic complexes uranium and thorium essentially utilize similar levels of 6d contributions, and the difference is that uranium uses more 5f whereas thorium uses less 5f and more 7s in the bonding. Interestingly, this is opposite to what is found for the analogous thorium– and uranium–phosphorus complexes [Th(Tren$^{TIPS}$)(PH$_2$)] (**III**), [U(Tren$^{TIPS}$)(PH$_2$)] (**IV**), [Th(Tren$^{TIPS}$)(PH)]$^−$ (**V**), [U(Tren$^{TIPS}$)(PH)]$^−$ (**VI**) (ref. 37), where similar 5f contributions were found and instead 6d participation was compromised by 7s contributions when those complexes are compared to the analogous uranium–phosphorus complexes overall (Table 2) (ref. 61). However, the thorium–arsenic linkages here are more polarized than the analogous thorium–phosphorus ones where comparisons are available[37]. Overall, these results certainly suggest a more nuanced bonding picture than that typically found, that is, a straight 5f versus 6d situation, and when phosphorus versus arsenic are considered provide evidence that the tension of 7s versus 6d versus 5f is also more complicated that might have previously been considered. Why the tension between 7s, 6d, and 5f orbitals varies between phosphorus and arsenic when bonded to thorium is currently unclear, and will require further isostructural series of compounds to be prepared and studied to be fully understood.

## Methods

**General.** Experiments were carried out under a dry, oxygen-free dinitrogen atmosphere using Schlenk-line and glove-box techniques. All solvents and reagents were rigorously dried and deoxygenated before use. All compounds were characterized by elemental analyses, NMR, FTIR, single crystal X-ray diffraction studies, and DFT, NBO, and QTAIM computational methods. The thorium–arsenic complexes reported here are on, or beyond, the cusp of stability and so their inherent instability coupled to their organosilyl-rich natures rendered the acquisition of reliable and meaningful microanalytical data problematic, as has been found elsewhere[76,77].

**Preparation of [Th(Tren$^{TIPS}$)(AsH$_2$)] (2).** DME (25 ml) was added to a cold (−78 °C) mixture of **1** (2.51 g, 2.0 mmol) and KAsH$_2$ (0.23 g, 2.0 mmol). The pale yellow slurry was allowed to warm to room temperature and stirred for 30 min to afford an orange solution. The solvent was removed *in vacuo* and the product was extracted into hexanes. Removal of hexanes *in vacuo* afforded a pale brown solid. Crystalline material was obtained from a hexanes solution (4 ml) stored at −30 °C for 24 h. Yield: 1.30 g, 71%. Anal. Calcd for C$_{33}$H$_{77}$AsN$_4$Si$_3$Th• 0.2C$_7$H$_8$: C, 43.97; H, 8.43; N, 5.96%. Found: C, 43.71; H, 8.81; N, 6.01%. $^1$H NMR (C$_6$D$_6$, 298 K): δ 1.24 (d, $^3J_{HH}$ = 7.21 Hz, 54H, CH(CH$_3$)$_3$), 1.40 (septet, $^3J_{HH}$ = 7.46 Hz,

9H, CH(CH$_3$)$_3$), 2.48 (t, $^3J_{HH}$ = 4.40 Hz, 6H, CH$_2$CH$_2$), 3.56 (t, $^3J_{HH}$ = 4.40 Hz, 6H, CH$_2$CH$_2$). As*H* resonance was not observed. $^{13}$C{$^1$H} NMR (C$_6$D$_6$, 298 K): δ 13.87 (CH(CH$_3$)$_2$), 20.29 (CH(CH$_3$)$_2$), 47.22 (CH$_2$), 64.19 (CH$_2$). $^{29}$Si{$^1$H} NMR (C$_6$D$_6$, 298 K): δ 5.76 (*Si*(CH(CH$_3$)$_2$)$_3$). FTIR (cm$^{−1}$): ṽ 2,939 (m), 2,888 (m), 2,862 (m), 2,064 (w), 2,051 (w), 459 (m), 1,272 (w), 1,260 (w), 1,105 (m), 1,040 (m), 925 (s), 880 (s), 805 (m), 728 (s), 671 (s), 630 (s), 558 (m), 515 (m). The synthesis of **2D** was accomplished by using KAsD$_2$ instead of KAsH$_2$.

**Preparation of [Th(Tren$^{TIPS}$)(μ-AsH)K(15C5)] (3).** Method A: a solution of 15-crown-5 (0.11 g, 0.5 mmol) in DME (25 ml) was added to a cold (−78 °C) mixture of **4** (0.422 g, 0.5 mmol) and KAsH$_2$ (0.058 g, 0.5 mmol). The yellow solution was allowed to warm to room temperature with stirring for a further 25 min. The solvent was removed *in vacuo* to afford a bright yellow solid. The solid was washed with pentane (2 × 5 ml) and dried *in vacuo* to afford a yellow powder. Method B: A solution of 15-crown-5 (24.0 mg, 0.108 mmol) in DME (15 ml) was added to a cold (−78 °C) mixture of **2** (99 mg, 0.108 mmol) and KCH$_2$Ph (14.2 mg, 0.108 mmol). The dark orange slurry was allowed to warm to room temperature with stirring for 15 min. The solvent was removed *in vacuo* to afford an orange oil. Crystalline material was obtained from a toluene solution (3 ml) stored at −30 °C for 24 h. Yield: 0.42 g, 70%. Anal. Calcd for C$_{33}$H$_{77}$AsN$_4$Si$_3$Th• 0.2C$_7$H$_8$: C, 43.78; H, 8.20; N, 4.75%. Found: C, 44.41; H, 7.92; N, 4.38%. $^1$H NMR (C$_6$D$_6$, 298 K): δ 1.55 (d, $^3J_{HH}$ = 7.46 Hz, 54H, CH(CH$_3$)$_3$), 2.05 (septet, $^3J_{HH}$ = 7.46 Hz, 9H, CH(CH$_3$)$_3$), 2.60 (t, 6H, CH$_2$CH$_2$), 3.06 (s, 16H, OCH$_2$), 3.70 (t, 6H, CH$_2$CH$_2$). As*H* resonance was not observed. $^{13}$C{$^1$H} NMR (C$_6$D$_6$, 298 K): δ 14.75 (CH(CH$_3$)$_2$), 21.08 (CH(CH$_3$)$_2$), 45.98 (CH$_2$), 65.33 (CH$_2$) 69.25(OCH$_2$). $^{29}$Si{$^1$H} NMR (C$_6$D$_6$, 298 K): δ 2.54 (*Si*(CH(CH$_3$)$_2$)$_3$). FTIR (cm$^{−1}$): ṽ 2,936 (m), 2,858 (w), 1,933 (w), 1,590 (w), 1,459 (m), 1,353 (m), 1,241 (s), 1,116 (s), 1,062 (s), 1,024 (s), 930 (s),881 (s), 804 (m), 740 (s), 670 (m), 626 (m), 511 (m). The synthesis of **3D** was accomplished by using KAsD$_2$ instead of KAsH$_2$.

**Preparation of [{Th(Tren$^{TIPS}$)}$_2$(μ-AsH)] (5).** DME (25 ml) was added to a cold (−78 °C) mixture of **4** (0.68 g, 0.8 mmol) and KAsH$_2$ (0.093 g, 0.8 mmol). The pale orange slurry was stirred at −78 °C for 30 min, allowed to warm to room temperature and stirred for a further hour to afford a pale orange solution. Solvent was removed *in vacuo* and the product was extracted into toluene to afford a dark orange solution. Removal of the solvent *in vacuo* afforded an orange powder. Crystalline material was obtained from a toluene solution (4 ml) stored at −30 °C for 24 h. Yield: 0.52 g, 74%. Anal. Calcd for C$_{66}$H$_{151}$AsN$_8$Si$_6$Th$_2$: C, 44.93; H, 8.63; N, 6.35%. Found: C, 43.23; H, 8.59; N, 5.16%. $^1$H NMR (C$_6$D$_6$, 298 K): δ 1.43 (d, $^3J_{HH}$ = 7.34 Hz, 54H, CH(CH$_3$)$_3$), 1.81 (septet, $^3J_{HH}$ = 7.34 Hz, 9H, CH(CH$_3$)$_3$), 2.53 (t, br, 6H, CH$_2$CH$_2$), 3.60 (t, 6H, CH$_2$CH$_2$). As*H* resonance was not observed. $^{13}$C{$^1$H} NMR (C$_6$D$_6$, 298 K): δ 14.51 (CH(CH$_3$)$_2$), 20.87 (CH(CH$_3$)$_2$), 45.95 (CH$_2$), 65.23 (CH$_2$). $^{29}$Si{$^1$H} NMR (C$_6$D$_6$, 298 K): δ 2.83 (*Si*(CH(CH$_3$)$_2$)$_3$). FTIR (cm$^{−1}$): ṽ 2,936 (m), 2,858 (m), 1,922 (w), 1,458 (m), 1,274 (w), 1,092 (m), 1,044 (m), 1,013 (m), 929 (s), 880 (s), 851 (m), 731 (s), 671 (s), 628 (s), 513 (m). The synthesis of **5D** was accomplished by using KAsD$_2$ instead of KAsH$_2$.

**Preparation of [{Th(Tren$^{TIPS}$)}$_2$(μ-As)][K(15C5)$_2$] (6).** A solution of 15-crown-5 (0.176 g, 0.8 mmol) and KAsH$_2$ (0.046 g, 0.4 mmol) in DME (20 ml) was added to a solution of **4** (0.68 g, 0.8 mmol) in DME (15 ml) dropwise over 15 min. The resulting yellow solution was stirred for a further 15 min and then the solvent was removed *in vacuo* to afford a bright orange solid. The product was washed with toluene (2 × 10 ml) to afford a bright orange powder. Crystalline material was obtained by dissolution in DME (5 ml) and storage at −30 °C for 24 h. Yield: 0.30 g, 74%. Anal. Calcd for C$_{86}$H$_{190}$AsN$_8$Si$_6$Th$_2$KO$_{10}$: C, 46.05;

H, 8.54; N, 5.00%. Found: C, 46.34; H, 8.56; N, 5.12%. $^1$H NMR (THF-$d_8$, 298 K): $\delta$ 1.27 (d, $^3J_{HH} = 6.60$ Hz, 108H, CH(CH$_3$)$_3$), 1.62 (septet, $^3J_{HH} = 6.62$ Hz, 18H, CH(CH$_3$)$_3$), 2.53 (t, br, 12H, CH$_2$CH$_2$), 3.64 (m, 52H, CH$_2$CH$_2$, OCH$_2$). $^{13}$C{$^1$H} NMR (THF-$d_8$, 298 K): $\delta$ 14.91 (CH(CH$_3$)$_2$), 21.63 (CH(CH$_3$)$_2$), 46.18 (CH$_2$), 59.08 (CH$_2$), 72.94 (OCH$_2$). $^{29}$Si{$^1$H} NMR (THF-$d_8$, 298 K): no resonance. FTIR (cm$^{-1}$): ṽ 2,933 (m), 2,857 (m), 1,460 (w), 1,355 (w), 1,247 (w), 1,119 (m), 1,051 (m), 933 (s), 880 (m), 794 (m), 738 (s), 671 (m), 625 (m), 555 (w), 514 (w).

**Data availability.** The X-ray crystallographic coordinates for structures reported in this article have been deposited at the Cambridge Crystallographic Data Centre (CCDC), under deposition number CCDC 1518196–1518199. These data can be obtained free of charge from the Cambridge Crystallographic Data Centre via www.ccdc.cam.ac.uk/data_request/cif. All other data are available from the corresponding authors on request.

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

## Acknowledgements

We thank the Royal Society (Grant UF110005), European Research Council (Grants 239621 and 612724), Engineering and Physical Sciences Research Council (Grants EP/G051763/1 and EP/M027015/1), Universities of Nottingham, Manchester, and Regensburg, the Deutsche Forschungsgemeinschaft, and COST Action CM1006 for generously supporting this work.

## Author contributions

E.P.W. and G.B. synthesized and characterized the compounds. A.J.W. carried out the single crystal X-ray diffraction analyses. S.T.L. performed and analysed the computational analyses, originated the central idea, and with M.S. supervised the work, analysed the data and wrote the manuscript with contributions from all the co-authors.

## Additional information

**Competing financial interests:** The authors declare no competing financial interests.

