## [Peer Review File · Nature Communications]

Reviewers' comments:

Reviewer #1 (Remarks to the Author):

Liddle and associates detail the synthesis of several thorium arsenic complexes featuring terminal AsH₂, bridging AsH and bridging As units. The compounds are well described and characterized by IR, NMR and X-ray diffraction. The electronic structure is analysed using standard DFT methods and NBO as well as AIM analyses. The research results are clearly presented and certainly of interest to the actinide community. In principle I am willing to recommend acceptance, however, I also believe that the authors should consider for future studies to avoid "unnecessary" fragmentation of their excellent experimental work. Previously they reported on an identical set of compounds, in which As is replaced by P (Ref. 37). Furthermore, nearly identical compounds have also been prepared for uranium as the metal centre. Nevertheless, returning to thorium I concede that these arsenic and P containing compounds are experimentally challenging, but why not write a full paper in which they directly compare these group 15 elements and their influence on the Th-E bond as inferred from DFT studies? However, this is just a minor comment and mostly represents this referee's preference for "complete" Articles over "less complete" Communications. Nevertheless, there are also some scientific points that need to be clarified before acceptance of the manuscript:

1. p. 4: "the presence of AsH₂ in 2 is confirmed by two As-H stretches at 2051 and 2064 cm⁻¹" However, when I consult the Supplementary material Figure 1, I can only detect one broad absorption. So why do the authors refer to two stretches? Next point which is also valid for Supplementary Figures 2 and 3. There is a clear change upon preparation of the D-labelled complexes 2D, 3D and 5D. In the manuscript the authors only mention that the resonances assigned to As-H shifts into the fingerprint regime upon deuteration. This is certainly true but there is also a clear change in this regime, i.e. one absorption gains intensity upon D-labelling. This stretch should be clearly assigned and mentioned accordingly. There is clear new stretch at ca. 1300 cm⁻¹ in Supplementary Figure S1. Ideally the authors will employ equally scaled IR spectra of labelled and unlabelled compounds and then form the difference spectra. I assume that this will be easy considering that they should have the electronic raw data in hand. I strongly believe that this would be beneficial for the paper.

Minor point: Figure captions for Supplementary Figures 2 and 3 are incorrect and should be corrected.

2. Did the authors entertain to record RAMAN spectra on 5? It may potentially be useful to gather some information on the strength of the Th-As bond, which is apparently strongly polarized.

3. What is the thermal stability of these complexes?

4. Some references on transition metal arsenic compounds may be added, e.g. Scherer (Acc. Chem. Res. 1999, 32, 751) and Scheer (Coord. Chem. Rev. 1997, 163, 271). Work by Cummins on (tren)Mo≡As (Chem Commun. 1998, 1777) and work by Schrock (JACS 1996, 118, 11305).

5. Would it be possible to include a table which features the computational results for isostructural U/Th-As and U/Th-P compounds (similar to their Table 1 in the manuscript)? This may make it a little easier to draw a direct comparison.

6. In their conclusions the authors argue that the simplistic picture 5f vs. 6d which is very frequently advocated by chemists in the actinide community may be too simplistic. This is an important point which makes this contribution particular valuable for the f-block community. They also state that in the case of P vs. As a "tension of 7s vs. 6d vs. 5f" makes this picture more complicated. Based on the computations I completely agree with this statement, but: "How robust are these computations when switching methods/functionals?" So far everything is based on computations and experimentally this information will be very hard to extract Although the

authors honestly admit that an explanation is hard to provide, but would it be possible to advance a model why this tension may exist for Th-As vs. Th-P?

Reviewer #2 (Remarks to the Author):

Liddle and co-workers report a series of thorium complexes with a variety of Th-As(H) bonding motifs in this article. This work is challenging both in practice and analysis. This chemistry represents the cutting edge of actinide/main group chemistry. That being said, the work in its present form is not publishable in this or any other journal. A review of the CIF shows the use of DFT calculations to restrain (very) poor refinements to match calculations. For starters, this is the opposite of how science is supposed to work. Theory is modified to fit data not vice versa. The resulting refinements are beyond the limits of credibility with restraints in one case totally 1802! The second terms of the weighting schemes are larger than 100 and in one case 159. The wR2 tops 20% in several structures. Even the space groups are worrisome. The first one is in Pna2(1), which is polar. This is important because few molecular systems adopt this space group. There are some, but normally the molecules have highly acentric shapes and intermolecular interactions (typically hydrogen bonds) create the polar c axis by aligning acentric shapes in a single direction. It is not clear to me how this happens in this system. The work from Liddle and co-workers is normally quite brilliant. This does not fit the mould from this group.

Reviewer #3 (Remarks to the Author):

This manuscript describes the synthesis, structure, and bonding of new thorium-arsenic compounds including the first actinide arsenide, which includes some multiple bonding character. Overall the manuscript is well-written and the compounds are well characterized (although compound 5 needs better combustion analysis), however this report is too close to the previously reported thorium and uranium phosphorus compounds to be published in Nature Communications. Absolutely beautiful compounds and unique to arsenic versus phosphorus but the approach to obtaining these compounds is too cookie-cutter.

Reviewers' comments:

Reviewer #1 (Remarks to the Author):

Little and associates detail the synthesis of several thorium arsenic complexes featuring terminal AsH₂, bridging AsH and bridging As units. The compounds are well described and characterized by IR, NMR and X-ray diffraction. The electronic structure is analysed using standard DFT methods and NBO as well as AIM analyses. The research results are clearly presented and certainly of interest to the actinide community. In principle I am willing to recommend acceptance, however, I also believe that the authors should consider for future studies to avoid “unnecessary” fragmentation of their excellent experimental work. Previously they reported on an identical set of compounds, in which As is replaced by P (Ref. 37). Furthermore, nearly identical compounds have also been prepared for uranium as the metal centre. Nevertheless, returning to thorium I concede that these arsenic and P containing compounds are experimentally challenging, but why not write a full paper in which they directly compare these group 15 elements and their influence on the Th-E bond as inferred from DFT studies? However, this is just a minor comment and mostly represents this referee’s preference for “complete” Articles over “less complete” Communications.

RESPONSE: We thank the reviewer for their positive support. Regarding ‘fragmentation’ we understand what the reviewer is saying but would like them to know that we do try to publish work when we feel it is complete and when, for example, our Th-P paper was submitted/refereed we had no idea whether the work in the present paper would work out. Further, the Th-P paper is 11 journal pages and our prior U-As paper is 9 journal pages so it’s hard to us to see how if we had waited then added this work in how we would have anything other than an unwieldy manuscript. We feel it is not unreasonable to say that the papers naturally define themselves by the U/Th/P/As series of combinations.

Nevertheless, there are also some scientific points that need to be clarified before acceptance of the manuscript:

1. p. 4: “the presence of AsH₂ in 2 is confirmed by two As-H stretches at 2051 and 2064 cm⁻¹” However, when I consult the Supplementary material Figure 1, I can only detect one broad absorption. So why do the authors refer to two stretches? Next point which is also valid for Supplementary Figures 2 and 3. There is a clear change upon

preparation of the D-labelled complexes 2D, 3D and 5D. In the manuscript the authors only mention that the resonances assigned to As-H shifts into the fingerprint regime upon deuteration. This is certainly true but there is also a clear change in this regime, i.e. one absorption gains intensity upon D-labelling. This stretch should be clearly assigned and mentioned accordingly. There is clear new stretch at ca. 1300 cm^{-1} in Supplementary Figure S1. Ideally the authors will employ equally scaled IR spectra of labelled and unlabelled compounds and then form the difference spectra. I assume that this will be easy considering that they should have the electronic raw data in hand. I strongly believe that this would be beneficial for the paper.

RESPONSE: When supplementary figure 1 is inspected in close detail it can just be made out that there are two overlapping absorptions, but we appreciate that this may well have been lost in the pdf conversion process. However, how well-resolved these two absorptions are varies from ATR-IR to ATR-IR. Therefore, we have re-recorded the spectrum of **2** where the two absorptions are better defined now, and hopefully this can be seen in the new SI file. Regarding the difference suggestion, this is in principle an excellent idea so we looked into this. However, there is an inherent issue to these systems that prevents this suggestion from being realised. The issue is that where heavy p-block elements are concerned multiple vibrational modes tend to end up coupling to the As-H vibrations in large molecules. Thus, the core vibrations in the fingerprint region and C-H region are affected and this does change significantly on changing H to D on As. Therefore, unfortunately but with good reason, the IR spectra of nH and nD are not directly subtractable. Those of us based in Manchester have noticed this since embarking on this An-E collaboration and those based in Regensburg have seen this phenomena many times over their extensive experience of heavy p-block chemistry spanning decades. The analytical frequencies calculations do not really pick up these nuances, and unfortunately the sheer size of these molecules prevents fuller vibrational analyses from being tractable computationally. That said, whilst this is an interesting issue we feel it is peripheral to the manuscript. This also accounts for the 1300 cm^{-1} absorbance for compound **2**, i.e. the different H/D coupled vibrations result in a sharper absorbance for the latter (so something has decoupled) – in this case the AsH₂ at $\sim 2064 \text{ cm}^{-1}$ would be predicted to move to ca 1474 cm^{-1} which already has moderately intense absorptions so cannot be observed.

Minor point: Figure captions for Supplementary Figures 2 and 3 are incorrect and should be corrected.

RESPONSE: Good spot and corrected.

2. Did the authors entertain to record RAMAN spectra on **5**? It may potentially be useful to gather some information on the strength of the Th-As bond, which is apparently strongly polarized.

RESPONSE: This is a good suggestion and we did look into this before submission. However, a frustrating and recurrent theme with our Tren(TIPS) complexes is their organosilicon-rich nature renders them incompatible with Raman. When we have the laser power high enough to expect a sample response they burn. When we turn the power down we get no response except sample warming. When we dilute the sample we get no response. This is frustrating for us as we use Raman on separate projects involving uranyl and get excellent responses, but here the Tren(TIPS) ligand has thwarted our efforts irrespective of what heteroatom sits in the pocket.

3. What is the thermal stability of these complexes?

RESPONSE: Samples can be handled at room temperature for short periods of time in solution but they tend to decompose. Solids in an inert atmosphere glove box are ok for days but they darken on extended storage.

4. Some references on transition metal arsenic compounds may be added, e.g. Scherer (*Acc. Chem. Res.* 1999, 32, 751) and Scheer (*Coord. Chem. Rev.* 1997, 163, 271). Work by Cummins on (tren)Mo \equiv As (*Chem Commun.* 1998, 1777) and work by Schrock (*JACS* 1996, 118, 11305).

RESPONSE: We have added a sentence at the end of paragraph 2 of the introduction highlighting that M-As multiple bonds for even transition metals are quite rare, providing an opportunity to introduce the suggested citations. We have added a number of reviews (Scheer, Scherer, Cummins) then also added some of the early results in the area from Schrock, Scheer, and Wolczanski (references 49-55 in revised manuscript). This takes the paper to 77 references which is ok editorially (10% overshoot usually allowed) but we could not then add further references, but by focusing on reviews the relevant literature is covered one way or another.

5. Would it be possible to include a table which features the computational results for isostructural U/Th-As and U/Th-P compounds (similar to their Table 1 in the manuscript)? This may make it a little easier to draw a direct comparison.

RESPONSE: This is a good suggestion so we have added a table (table 2) compiling the relevant NBO data and

modified the discussion section to reflect its inclusion.

6. In their conclusions the authors argue that the simplistic picture 5f vs. 6d which is very frequently advocated by chemists in the actinide community may be too simplistic. This is an important point which makes this contribution particular valuable for the f-block community. They also state that in the case of P vs. As a “tension of 7s vs. 6d vs. 5f” makes this picture more complicated. Based on the computations I completely agree with this statement, but: “How robust are these computations when switching methods/functionals?” So far everything is based on computations and experimentally this information will be very hard to extract Although the authors honestly admit that an explanation is hard to provide, but would it be possible to advance a model why this tension may exist for Th-As vs. Th-P?

RESPONSE: The referee raises a very interesting point. We've always strived to point out that of course the results have to be viewed as qualitative rather than quantitative. One thing we would say though is that the calculations are internally consistent. We've done a lot of calculations in the past 9 years (>150 easily) all at exactly the same level of theory in terms of functional and basis sets. In addition, STL has overseen them all himself to ensure absolute internal consistency. As a researcher whose PhD was strongly crystallographic, STL has long been aware of how variability of treatments in modeling can be problematic and when independent and embarking on DFT calculations STL decided such work needed a consistent treatment which is why he oversees them himself. So the trend will be consistent even if the magnitudes are perhaps open to debate. It is tempting to speculate on the f vs d vs s split, and we would like to be able to, but we stand by our final statement that more work really needs to be done to fully tease this out.

Reviewer #2 (Remarks to the Author):

Liddle and co-workers report a series of thorium complexes with a variety of Th-As(H) bonding motifs in this article. This work is challenging both in practice and analysis. This chemistry represents the cutting edge of actinide/main group chemistry.

RESPONSE: We thank the reviewer for acknowledging the importance of the work.

That being said, the work in its present form is not publishable in this or any other journal. A review of the CIF shows the use of DFT calculations to restrain (very) poor refinements to match calculations. For starters, this is the opposite of how science is supposed to work. Theory is modified to fit data not vice versa.

RESPONSE: We believe there is a misunderstanding here because our initial text was not complete and clear, so we apologise for the lack of clarity. The As-H were initially located in the difference maps. However, mindful that we were refining one or two H atoms next to Th-As units we restrained them. The DFT was used as simply a guide and this was coupled to a survey of the literature to ensure that sensible bond lengths and angles were used, based on the initial locations from the difference map. Since they are crystallographic restraints and not constraints the models are not totally rigid. In many regards this is no different to the riding models used to refine C-H positions, which is standard nowadays. Indeed, we would suggest that the treatment of the As-Hs is better due to libration effects for C-H bonds but everyone fixes the latter in riding models with no question even though they usually constitute the majority of atoms in structures. This method of E-H treatment has also been previously accepted after multiple rounds of review including by specialist crystallographic reviewers called in specifically to comment on this aspect both at Nature Chemistry and this journal. Clarifying comments have been added to the cif file, where we realise the initial step, that is location in the difference map, had been missed out before from the refinement_special_details sections.

The resulting refinements are beyond the limits of credibility with restraints in one case totally 1802!

RESPONSE: We are a bit puzzled by this comment because the number of restraints referred to simply reflects the nature of the disorder. In compound **2** two entire SiPr_3 groups are disordered each over two locations and each in approximate 0.6:0.4 ratios. So there are in total 62 C, H, and Si atoms split over 124 positions in the presence of Th. This requires restraints to model 1,2- and 1,3-distances as well as rigid bond restraints and thermal motion. This, taken together with other restraints, e.g. the As-H bonds, gets you quite quickly to this range of restraints. Although it would always be desirable for the data to be better in practice they aren't going to be since it has been a real struggle to get good crystals of **2** so this is the best data we could ever get on this compound, but the connectivity and identity is clear-cut. We end up with a publishable set of data because the most important bond in the structure, Th-As, is the best determined being between the two heaviest atoms in the structure, with a bond length determined to three decimal places and a su on the third dp in low single figures, which is fine. The structure has a data to parameter ratio of 12.5:1 that is more than enough, the model is well-behaved with essentially no shifts in the final refinement and ellipsoids are fine. Although it is always possible to focus on various crystallographic metrics the data are publishable, confirm the

compound identity, and yield the bond length of most importance to a good level of precision. We have also re-refined the other structures, see below.

The second terms of the weighting schemes are larger than 100 and in one case 159.

RESPONSE: While the weighting schemes found in the refinement of our reported structures are high based on classical standards, they are not due to any un-modelled disorder or twinning and are merely a genuine reflection of the data obtained. The GoF numbers are also rather good and the refinements are settled.

The wR2 tops 20% in several structures.

RESPONSE: Only one compound had a wR2>20% and since submission we have obtained a better dataset for compound **5**, (new wR2 = 8% and 23 restraints rather than wR2 = 20% with 1620 restraints) and have re-refined the datasets for compounds **3** and **6** to reduce the number of restraints present, however due to the multiple sets of disordered components within both structures restraints were still required to give chemically sensible models; previously: **3**, 1502; **6**, 1202. New: **3**, 873; **6**, 689. We believe that the remaining restraints used are justified and make chemical sense. We would also point out that by definition wR2 is usually 2-3 times larger than R1 values, which are perfectly reasonable in the new cif files supplied (R1 = 10.5, 6.3, 4.3, and 7.2% for **2**, **3**, **5**, and **6**, respectively). The data:parameter ratios are also very acceptable: **2**, 7252:581; **3**, 14715:691; **5**, 18521:798; **6**, 20018:1092.

Even the space groups are worrisome. The first one is in Pna2(1), which is polar. This is important because few molecular systems adopt this space group. There are some, but normally the molecules have highly acentric shapes and intermolecular interactions (typically hydrogen bonds) create the polar c axis by aligning acentric shapes in a single direction. It is not clear to me how this happens in this system. The work from Liddle and co-workers is normally quite brilliant. This does not fit the mould from this group.

RESPONSE: A search of the CCDC (Based on a search of the CSD database 16th December 2016, *Acta Cryst.*, **2016**, B72, 171-179) finds that there are 7342 examples of structures published in this space group (or its equivalents), so it is not particularly unusual. That said, there is some pseudo-symmetry present in the dataset for **2** (See appendix 1 for Space group determination output for Compound **2**). While the c glide on the short axis is clear, the n glide on the intermediate axis is less clear, but we believe it to be present. Although screw axes are more difficult to find than glide planes, considering the condition for observed intensity for the 2₁ screw on the b axis (k=2n) does not fully overlap with the conditions required for the c (l=2n) or n (h+k= 2n) glides we believe there to be a 2₁ screw axis present on the long axis. Putting the assigned space group of Pc2₁n into a standard setting leads to the space group assignment of Pna2₁. If the assignment of a 2₁ screw axis is incorrect, the alternate space group assignment would be the centrosymmetric space group Pnam (standard setting Pnma). While the structure can be solved in this space group the mirror plane runs through the N_{amine}-Th-As plane, with the whole molecule being disordered over the mirror plane. Refinement of this did not lead to a more sensible model. Due to the pseudo-symmetry and large amount of disorder present, restraints were required to provide a satisfactory refinement of **2**, however we believe the structure and connectivity to be unambiguous. The other space groups (Pbcn, P-1, P2(1)/c) seem perfectly reasonable to us also.

Reviewer #3 (Remarks to the Author):

This manuscript describes the synthesis, structure, and bonding of new thorium-arsenic compounds including the first actinide arsenide, which includes some multiple bonding character. Overall the manuscript is well-written and the compounds are well characterized (although compound 5 needs better combustion analysis), however this report is too close to the previously reported thorium and uranium phosphorus compounds to be published in Nature Communications. Absolutely beautiful compounds and unique to arsenic versus phosphorus but the approach to obtaining these compounds is too cookie-cutter.

RESPONSE: With respect to this referee we disagree with this report. As they even themselves point out this paper contains a number of novel 'firsts' and they effectively acknowledge that phosphorus is not arsenic, and thorium is certainly not uranium. If the work were 'cookie-cutter' then it would be an exact match to prior work but this is not the case; experimentally we show we've gone right to the edge of stability with these systems. We therefore feel this work represents an important advance, especially when the new orbital tensioning contrast of P vs As issue emerges as it does.

Appendix 1

```
*****  
* GRAL PLUGIN started at Fri Dec 16 09:26:26 2016 *  
* version 2.3.3 *  
*****
```

38.41 (release 15-09-2015), compiled Sep 15 2015, 16:02:48

original cell in Angstroms and degrees:

12.607500 16.210560 20.829110 89.9885 89.9865 89.9888

36020 Reflections read from file ast136.hkl

29617 Reflections used for space-group determination (up to diffraction limit of 0.85Å); mean (I/sigma) = 5.14

Lattice exceptions: P A B C I F Obv Rev All

N (total) = 0 14759 14812 14835 14857 22203 19733 19734 29617

N (int>3sigma) = 0 8187 8083 8162 8197 12216 10932 10920 16402

Mean intensity = 0.0 61.4 46.6 60.8 61.3 56.2 61.7 61.7 61.3

Mean int/sigma = 0.0 5.2 4.5 5.1 5.1 4.9 5.1 5.1 5.1

Lattice type: B chosen Volume: 4256.94

DETERMINATION OF REDUCED (NIGGLI) CELL

Transformation from original cell (HKL F-matrix):

-0.5000 0.0000 0.5000 0.5000 0.0000 0.5000 0.0000 1.0000 0.0000

Unitcell: 12.172 12.175 16.211 89.98 90.00 62.37

Niggli form: a.a = 148.169 b.b = 148.231 c.c = 262.782
b.c = 0.054 a.c = 0.014 a.b = 68.726

SEARCH FOR HIGHER METRIC SYMMETRY

Option: [13] err= 0.084 ORTHORHOMBIC C-lattice R(int) = 0.071 [12766] vol = 4256.9
Cell: 12.608 20.829 16.211 89.99 89.99 89.99 Volume: 4256.94
Matrix: -1.0000 0.0000 0.0000 0.0000 0.0000 -1.0000 0.0000 -1.0000 0.0000

Option: [10] err= 0.075 MONOCLINIC C-lattice R(int) = 0.066 [10996] vol = 2128.5
Cell: 20.829 12.608 16.211 89.99 90.01 90.01 Volume: 4256.94
Matrix: 0.0000 0.0000 1.0000 -1.0000 0.0000 0.0000 0.0000 -1.0000 0.0000

Option: [14] err= 0.093 MONOCLINIC C-lattice R(int) = 0.065 [11064] vol = 2128.5
Cell: 12.608 20.829 16.211 90.01 90.01 89.99 Volume: 4256.94
Matrix: 1.0000 0.0000 0.0000 0.0000 0.0000 1.0000 0.0000 -1.0000 0.0000

Option: [34] err= 0.056 MONOCLINIC P-lattice R(int) = 0.066 [11029] vol = 2128.5
Cell: 12.172 16.211 12.175 89.98 117.63 90.00 Volume: 2128.47
Matrix: 0.5000 0.0000 -0.5000 0.0000 1.0000 0.0000 0.5000 0.0000 0.5000

Option: [31] err= 0.000 TRICLINIC P-lattice R(int) = 0.053 [7559] vol = 2128.5
Cell: 12.172 12.175 16.211 89.98 90.00 62.37 Volume: 2128.47
Matrix: -0.5000 0.0000 0.5000 0.5000 0.0000 0.5000 0.0000 1.0000 0.0000

option [13] selected

SPACE GROUP DETERMINATION

Lattice exceptions: P A B C I F Obv Rev All

N (total) = 0 14759 14835 14812 14857 22203 19733 19749 29617
 N (int>3sigma) = 0 8187 8162 8083 8197 12216 10932 10920 16402
 Mean intensity = 0.0 61.4 60.8 46.6 61.3 56.2 61.7 61.9 61.3
 Mean int/sigma = 0.0 5.2 5.1 4.5 5.1 4.9 5.1 5.2 5.1

Crystal system ORTHORHOMBIC and Lattice type P selected

Mean |E*E-1| = 0.768 [expected .968 centrosym and .736 non-centrosym]

Systematic absence exceptions:

	21--	b--	c--	n--	-21-	-a-	-c-	-n-	--21	--a	--b	--n
N	18	869	861	872	16	506	519	517	26	502	505	483
N I>3s	11	367	0	367	0	347	321	314	0	308	310	166
	21.0	77.5	0.4	77.2	2.6	61.9	82.2	80.2	1.5	144.8	144.6	14.5
<I/s>	3.6	5.2	0.2	5.2	1.2	6.1	6.8	6.7	0.7	8.2	8.3	2.3

Space Group	No.	C/A	En.	O.A.	Pie.	Pyr.	CCDC	ICSD	R(int)	N(eq)
Pmc2(1) (ba-c)	26	A	N	?	Y	Y	55	65	0.076	21602
Pma2 (cab)	28	A	N	?	Y	Y	5	25	0.076	21498
Pmma (-cba)	51	C	N	N	N	N	18	90	0.082	24882

Space Group	No.	C/A	En.	O.A.	Pie.	Pyr.	CCDC	ICSD	R(int)	N(eq)
Pnc2 (ba-c)	30	A	N	?	Y	Y	40	15	?	0
Pmna (-cba)	53	C	N	N	N	N	45	46	?	0
Pcc2 (abc)	27	A	N	?	Y	Y	8	3	?	0
Pccm (abc)	49	C	N	N	N	N	5	4	?	0
Pca2(1) (abc)	29	A	N	?	Y	Y	1665	177	?	0
Pbcm (ba-c)	57	C	N	N	N	N	307	197	?	0
Pmc2(1) (ba-c)	26	A	N	?	Y	Y	55	65	?	0
Pma2 (cab)	28	A	N	?	Y	Y	5	25	?	0
Pmma (-cba)	51	C	N	N	N	N	18	90	?	0

Space Group	No.	C/A	En.	O.A.	Pie.	Pyr.	CCDC	ICSD	R(int)	N(eq)
Pnna (-cba)	52	C	N	N	N	N	185	100	?	0
Pbcn (a-cb)	60	C	N	N	N	N	2065	409	?	0
Pban (a-cb)	50	C	N	N	N	N	18	12	?	0
Pnc2 (ba-c)	30	A	N	?	Y	Y	40	15	?	0
Pmna (-cba)	53	C	N	N	N	N	45	46	?	0
Pccn (abc)	56	C	N	N	N	N	806	83	?	0
Pcca (ba-c)	54	C	N	N	N	N	92	42	?	0
Pcca (abc)	54	C	N	N	N	N	92	42	?	0
Pcc2 (abc)	27	A	N	?	Y	Y	8	3	?	0
Pccm (abc)	49	C	N	N	N	N	5	4	?	0
Pbcn (ba-c)	60	C	N	N	N	N	2065	409	?	0
Pbca (a-cb)	61	C	N	N	N	N	8652	649	?	0
Pcca (-cba)	54	C	N	N	N	N	92	42	?	0
Pca2(1) (abc)	29	A	N	?	Y	Y	1665	177	?	0
Pbcm (ba-c)	57	C	N	N	N	N	307	197	?	0
Pna2(1) (cab)	33	A	N	?	Y	Y	3468	651	?	0
Pnma (-cba)	62	C	N	N	N	N	3416	4336	?	0
Pca2(1) (cab)	29	A	N	?	Y	Y	1665	177	?	0
Pbcm (a-cb)	57	C	N	N	N	N	307	197	?	0
Pba2 (cab)	32	A	N	?	Y	Y	48	26	?	0

REVIEWERS' COMMENTS:

Reviewer #1 (Remarks to the Author):

I want to thank the authors for their very detailed reply to the question/comments that I raised in my original review.

In my opinion their answers regarding comments on additional vibrational studies were adequately addressed and the requested additions to the computational section and references were included. Overall, this referee believes that the manuscript is now suitable for publication.

It will be interesting to see if the authors can also extend their investigations to the even more challenging Sb-derivatives.

Reviewer #2 (Remarks to the Author):

I am mostly satisfied with the responses to my comments. Indeed there was confusion concerning the use of constraints versus restraints in this work, and the authors have handled this correctly. There is a big differences between the two. I do object, however, to simply doing a search for the number of occurrences of Pna2(1) as a rebuttal the rarity of a given space group. CCSD now incorporates purely inorganic structures, so this number will be skewed by oxoanion compounds with tetrahedral and C3v moieties that lend themselves to the formation of noncentrosymmetric/polar structures. Second, I have not checked the total number of structures recently, but last time I did it was many hundreds of thousands. My point was not that it does not occur (we've made many compounds in this space group), but rather in molecular compounds their needs to be a good reason for it to happen. The authors analysis of the packing suffices. I am requesting no further revisions and appreciate the clarity added to the manuscript.

REVIEWERS' COMMENTS:

Reviewer #1 (Remarks to the Author):

I want to thank the authors for their very detailed reply to the question/comments that I raised in my original review. In my opinion their answers regarding comments on additional vibrational studies were adequately addressed and the requested additions to the computational section and references were included. Overall, this referee believes that the manuscript is now suitable for publication. It will be interesting to see if the authors can also extend their investigations to the even more challenging Sb-derivatives.

RESPONSE: We thank the reviewer for their constructive suggestions throughout which have helped us to improve the paper.

Reviewer #2 (Remarks to the Author):

I am mostly satisfied with the responses to my comments. Indeed there was confusion concerning the use of constraints versus restraints in this work, and the authors have handled this correctly. There is a big differences between the two. I do object, however, to simply doing a search for the number of occurrences of Pna2(1) as a rebuttal the rarity of a given space group. CCSD now incorporates purely inorganic structures, so this number will be skewed by oxoanion compounds with tetrahedral and C3v moieties that lend themselves to the formation of noncentrosymmetric/polar structures. Second, I have not checked the total number of structures recently, but last time I did it was many hundreds of thousands. My point was not that it does not occur (we've made many compounds in this space group), but rather in molecular compounds their needs to be a good reason for it to happen. The authors analysis of the packing suffices. I am requesting no further revisions and appreciate the clarity added to the manuscript.

RESPONSE: We thank the reviewer for helping us to improve the paper through their scrutiny.